# Numerical Modeling of Thermal Behavior during Lunar Soil Drilling

**Deming Zhao [1]**, **Zhisheng Cheng [1]**, **Weiwei Zhang [2],***, **Jinsheng Cui [3]** and **He Wang [1]**

1   School of Mechanical Engineering, Zhejiang Sci-Tech University, Hangzhou 310018, China
2   School of Mechanical Engineering, Harbin Institute of Technology, Harbin 150001, China
3   School of Mechanical and Electrical Engineering, Guangzhou University, Guangzhou 510006, China
*   Correspondence: zweier@hit.edu.cn

**Abstract:** This paper presents a detailed thermal simulation analysis of the drilling process for icy soil in the lunar polar region. The aim is to investigate the temperature changes that occur in the debris removal area during the drilling process. We developed a multi-level particle size simulation model that includes a thermal sieve based on geometric constraints to evaluate the influence of specific heat capacity and thermal conductivity on particle temperature. Using the central composite design method, we carried out the simulation test design and analyzed the average temperature difference of particles within and outside the range of the thermal sieve. The parameters of the discrete element model were determined by comparing the temperature of the debris removal zone in the lunar environment with the temperature simulated by the discrete element method. The results show that the thermal conductivity of the sieve ranges from 100 to 400 W/m, and the average temperature inside the thermal sieve is negatively related to the specific heat capacity. The temperature deviation of the chip removal area is ±10 °C, which is consistent with the temperature deviation observed in the lunar environment and the lunar icy regolith drilling test. Furthermore, the addition of the thermal sieve to the multi-stage particle size simulation modeling significantly reduces the calculation time by 86%. This reduction in computational time may potentially increase the efficiency of drilling operations in the future. Our study provides insights into the thermal behavior of lunar icy regolith during drilling, and proposes a numerical model of heat transfer with a thermal sieve that can effectively reduce computational time while ensuring accurate temperature calculations.

**Keywords:** lunar icy regolith drilling; thermal sieve; multi-level particle size simulation model; thermal simulation analysis





## 1. Introduction

At present, human geological exploration of the moon is concentrated in the low-latitude illumination area, and research on the permanent shadow area of the Moon's polar regions has not yet been conducted. However, through theoretical analysis and exploration results, scientists predict that the subsurface of lunar soil in the permanent shadow area of the Moon may contain water ice, methane, hydrogen sulfide, and other substances that could serve as resources for survival and energy supply [1]. The distribution and state of lunar icy regolith is still unknown, so detecting water ice is an important task of geological exploration in the permanent shadow area. According to the phase change characteristics of water, the phase change between gas and solid occurs at around −70 °C to −60 °C in a vacuum environment [2]. During the process of drilling and sampling, the drilling tool generates heat, which can cause the temperature to exceed the phase change temperature of water ice and cause sublimation, thereby damaging the scientific information in the sample. Therefore, studying temperature change in lunar soil is crucial for the collection and analysis of water ice in the permanent shadow area.

The discrete element method (DEM) was first proposed by Professor Cundall P.A. in 1971 [3,4] for rock mechanics, and later expanded to soil mechanics by Cundall P.A. and

Strack O.D.L. in 1979 [5–8]. Currently, some scholars use the DEM to simulate heat conduction in granular materials. Vargas et al. introduced the contact heat transfer model into the DEM and named it the thermal particle dynamic method, which was used to numerically analyze the heat transfer problem of particle [9–11]. The numerical results show that the DEM can effectively simulate heat transfer in particles and accurately reflect the overall heat transfer characteristics of the particles. Zhang and Zhou proposed an equivalent averaging technique to study the effective thermal conductivity of particle systems. They obtained the average heat flow and temperature gradient of the particle system in the DEM and calculated the effective thermal conductivity of the particle system. They also compared their results with the finite element method and studied the effects of particle diameter, solid volume fraction and coordination number on the thermal conductivity of the particle system [12,13].

Using the DEM, the research found that the heat affected area of the lunar soil particles is 40 mm when drilling 100 mm in the simulated lunar icy regolith. To consider the heat affected area under the condition of limited computational power, the study used a multi-level particle size simulation model by Cui Jinsheng to conduct the DEM of the lunar icy regolith. However, the multi-stage particle size simulation model will mix particles of different particle sizes in the process of drilling, resulting in inconsistent chip removal and temperature changes with the actual experiment. To address this issue, the research proposes a thermal sieve that prevents particles from passing through but does not affect the heat transfer of particles inside and outside the sieve. The sieve is used to limit the mutual flow of particles of different sizes and ensure the accuracy of thermal simulation analysis of the lunar icy regolith under the condition of limited computational power. In this study, we investigated the impact of the thermal parameters of the thermal sieve on particle heat transfer, and determined the optimal thermal sieve parameters. Our research also presents a modeling approach for discrete element thermal simulation that can effectively simulate large-scale particle flows.

## 2. Thermal Simulation Model

### 2.1. Particle Heat Conduction Model

The heat transfer process between particle media during drilling involves three mechanisms: heat conduction, convection, and radiation. These mechanisms include heat transfer inside particles, heat conduction between two particles through contact, convective heat transfer through fluid, and radiative heat transfer between particle surfaces. However, heat conduction between particles generally plays the dominant role. Therefore, the comprehensive heat transfer effect of the particle system is typically equivalent to the heat conduction between particles, which is referred to as effective heat conduction. This is characterized by effective thermal conductivity (ETC) and simplifies the heat transfer model.

In the current discrete element method, several scholars have studied the calculation model for heat transfer between particles [14]. The heat transfer between two particles (*i* and *j*) and between the particles and the drilling tool is calculated using the following formula:

$$\begin{cases} Q_{ij} = 2k_s \left( \frac{3F_n r^*}{4E^*} \right)^{1/3} (T_j - T_i) \\ E^* = \left( \frac{1-v_i^2}{E_i} + \frac{1-v_j^2}{E_j} \right)^{-1} \\ r^* = \frac{r_i r_j}{r_i + r_j} \end{cases} \tag{1}$$

where $Q_{ij}$ is the heat transfer rate from particle *j* to particle *i*, $k_s$ is the thermal conductivity of the granular materials, $F_n$ is the normal contact force between particles, $E$ is the modulus of elasticity, $v$ is the Poisson's ratio, $r$ is the contact radius, and $T$ is the particle temperature.

The model presented only considers heat transfer between particles, and does not take into account convection and radiation heat transfer. However, other forms of heat transfer can be approximated to the effective heat transfer between particles, which is

represented by the ETC, denoted as $k_s$ [15]. This model simplifies the heat transfer process. When the particles have a small diameter, the heat transfer inside the particles can be considered instantaneous, and the temperature distribution within the particle is assumed to be uniform.

When calculating the heat transfer between the drilling tool and the particles, the diameter of the drilling tool is considered infinite, because it is much larger than the diameter of the particles.

Based on the heat transfer relationship between two particles, if particle *i* contacts n particles, the temperature change of particle *i* can be calculated using the following equation:

$$\frac{dT_i}{dt} = \frac{\sum_{j=1}^{n} Q_{ij}}{\rho_i c_i V_i} \tag{2}$$

where *t* is the time, *n* is the number of particles in contact, $\rho_i$ is the density of particle *i*, $c_i$ is the specific heat capacity of particle *i*, and $V_i$ is the volume of particle *i*.

The focus of this paper is to investigate the impact of thermal sieves on particle heat transfer; thus, the heat transfer inside a single thermal sieve is not considered. In practice, thermal sieves are segmented and discrete. It is assumed that the internal temperature of each section of the thermal sieve is uniformly distributed.

### 2.2. Assumptions

In order to simplify the calculation, the following assumptions are made for this mode:

1. Because most temperatures do not exceed the sublimation temperature of water ice during drilling, the gaseous water generated can be ignored. Therefore, it is assumed that heat conduction plays a significant role, and effective heat conduction between particles is considered equivalent to all of the heat transfer in a particle system;
2. The simulated lunar soil is assumed to be composed of spherical particles;
3. The temperature inside each individual particle is assumed to be uniformly distributed;
4. The internal temperature of the thermal sieve is also assumed to be uniform;
5. The temperature of the particles and the geometry remain fixed during each simulation time step.

### 2.3. Simulation Model

2.3.1. Particle Parameters of Simulation Model

The Hertz–Mindlin model is one of the most fundamental contact models, and is often used as the basis for coupling different contact models [7,16]. In this paper, the Hertz–Mindlin model and parallel bonding model [17] are coupled to simulate the contact force of the lunar soil particles. The Hertz–Mindlin model defines the relationship between force and contact between particles, and between particles and geometry. The parallel bonding model was added to simulate particle bonding due to water ice. The principle of the parallel bond model is to add a bond between the particles in contact. The role of this bond is to form a whole of the particles in contact, similar to the lunar icy regolith. When the particle is subjected to an external force that reaches the endurance limit of the bond, the bond breaks. The parameter settings of the simulated lunar soil particles are shown in Tables 1 and 2, and the parameters has been verified by relevant scholars to meet the requirements of the simulation model of the lunar icy regolith [15,18,19].

**Table 1.** Particle parameters of simulated lunar soil.

| Parameter | Numerical Value |
| --- | --- |
| Particle thermal conductivity (W/m·K) | 14.76 |
| Specific heat capacity of particles (J/kg °C) | 800 |
| Particle diameter (mm) | First region 0.6, Second region 2.4 |

**Table 1.** *Cont.*

| Parameter | Numerical Value |
|---|---|
| Particle density (kg/m$^3$) | $3 \times 10^3$ |
| Particle shear modulus (Pa) | $3 \times 10^9$ |
| Particle Poisson's ratio | 0.25 |
| Particle recovery coefficient | 0.24 |
| Particle–particle static friction coefficient | 0.8 |
| Particle–particle rolling friction coefficient | 0.6 |

**Table 2.** Basic Parameters of Parallel Bonding Model.

| Parameter | Numerical Value |
|---|---|
| Normal stiffness per unit area | $3 \times 10^{10}$ N/m$^3$ |
| Normal range | 0 N/m$^3$ |
| Tangential stiffness per unit area | $1.5 \times 10^9$ m$^3$ |
| Tangential range | 0 N/m$^3$ |
| Normal strength | $1.2 \times 10^{11}$ Pa |
| Shear strength | $6 \times 10^{10}$ Pa |
| Bonded disk scale | 1 |

### 2.3.2. Simulation Model

Cui Jinsheng, from Guangzhou University, conducted a study on the thermal characteristics and temperature distribution of simulated lunar soil under atmospheric pressure and vacuum conditions, using the DEM [15,20]. However, the large number of simulated particles required for establishing the simulated lunar soil lead to a very long simulation time. To overcome this problem and ensure a certain level of simulation accuracy, Cui proposed a multi-level particle size simulation model, as shown in Figure 1a. The variable particle diameter refers to the large variation in particle diameter in different regions. The basic idea is to divide the simulated lunar soil into several regions. The first region is the region directly interacting with the drilling tool and its vicinity. The remaining simulated lunar soil is then divided into two or three regions from the inside to the outside. The simulated lunar soil in these regions mainly acts as a boundary condition, and is subject to small stress and weak mobility. Because the particles in the first region come into direct contact with the drilling tool, the temperature increase is greater during the drilling process. Therefore, filling the first region with small particles may better reflect the temperature changes during the drilling process. The particles in the second region are far away from the drilling tool and have a lesser temperature increase. The heat transfer in this region is mainly related to the thermal conductivity of particles, and is almost independent of particle size. Therefore, using large particles in this region can reduce the number of particles. This not only reduces the amount of calculation and saves the calculation time, but also does not significantly affect the simulation accuracy.

In this study, thermal sieves are added between the regions of the secondary particle size simulation model, as shown in Figure 1b. The thermal sieve is a surface structure that limits the mutual flow of particles between different regions, without affecting the heat transfer of particles between regions. After adding the thermal sieves, the particles in the two regions are separated and do not come into contact with each other. Therefore, the thermal sieves need to have a certain degree of thermal conductivity to ensure the normal transfer of heat. In this simulation model, a total of 10 thermal sieves with a radius of 11 mm and a length of 12 mm are set at the junction of the first and second regions, and the temperature of each thermal sieve is uniform. Thermal sieves 1–9 are cylindrical surfaces without an upper and lower surface, and thermal sieve 10 is a cylindrical surface without an upper surface, as shown in Figure 1b. The lunar soil barrel is set as a cylinder with a diameter of 70 mm and a length of 150 mm. The first area surrounded by the thermal sieve is 11 mm in radius and 120 mm in length. The inside of the particle sieve is filled with small

particles to simulate the lunar regolith removal effect. Outside the particle sieve, in order to achieve a higher computational efficiency, large particles are used for filling.

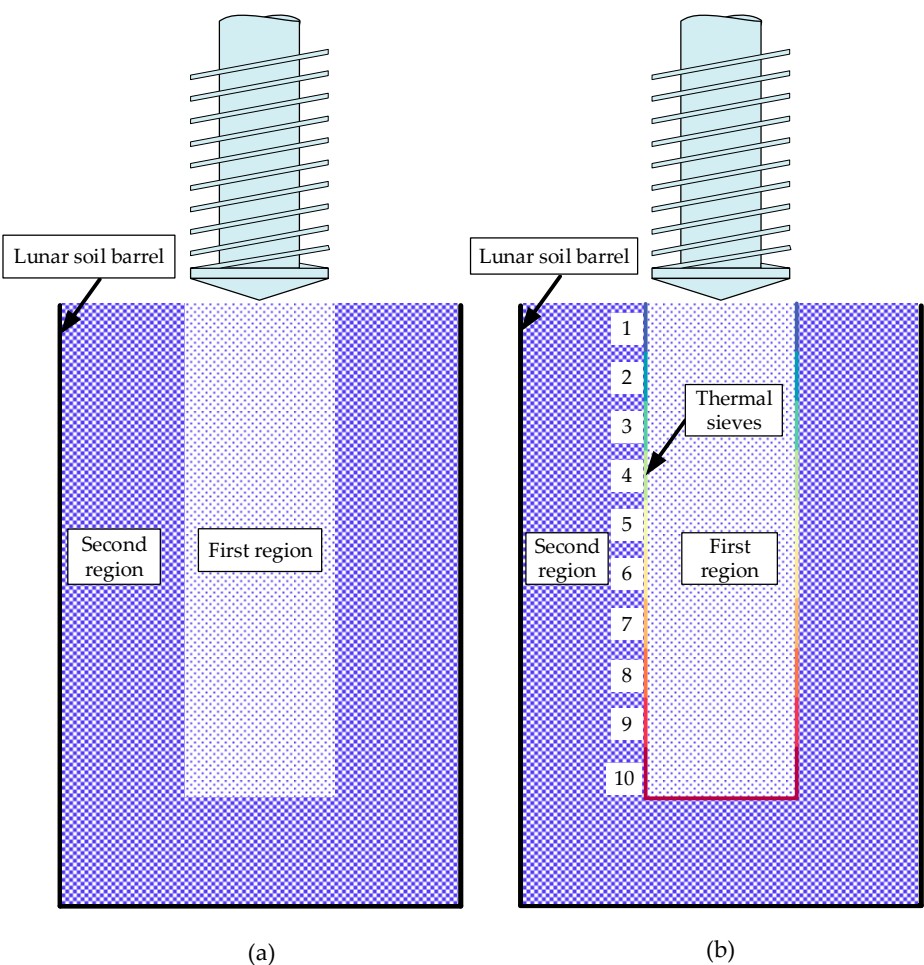

(a) (b)

**Figure 1.** Multi-level particle size simulation model; (**a**) Without thermal sieve; (**b**) Added thermal sieve.

## 3. Simulation Matching Test Design

*3.1. Simulation Matching Test Model*

In this study, Formulas (1) and (2) are used to calculate the heat transfer between the thermal sieve and the particles. The main parameters that affect the heat transfer of the particle system include the thermal conductivity and specific heat capacity of the thermal sieve, the elastic modulus and diameter and density of the sieve, and the initial temperature of the sieve. The elastic modulus of the thermal sieve is determined by the material's shear modulus and Poisson's ratio. To ensure consistency with the simulated lunar soil particles, the numerical value is selected accordingly. The density of the sieve has little effect on heat transfer and is therefore not matched. The diameter of the thermal sieve mainly depends on the range of the first area, and the diameter of the thermal sieve is much larger than the particle diameter, so it is not matched. To maintain the internal temperature balance of the simulated lunar soil without drilling, the initial temperature of the thermal sieve is set to the initial temperature of the particles. Hence, the thermal conductivity and specific heat capacity of the sieve are the matching parameters in this simulation.

To analyze the influence of the thermal conductivity and specific heat capacity of the thermal sieve on particle heat transfer, a parameter-matching simulation model is established, as shown in Figure 2. The model consists of particles with a diameter of 0.6 mm, placed in an insulated cylinder with a diameter and length of 30 mm. The front plate of the cylinder is set as the heat source, with a temperature of 700 °C, while the cylinder temperature is set at −196 °C and insulated. The thermal sieve is placed

10 mm away from the front plate, with an initial temperature of −196°C for both the sieve and particles.

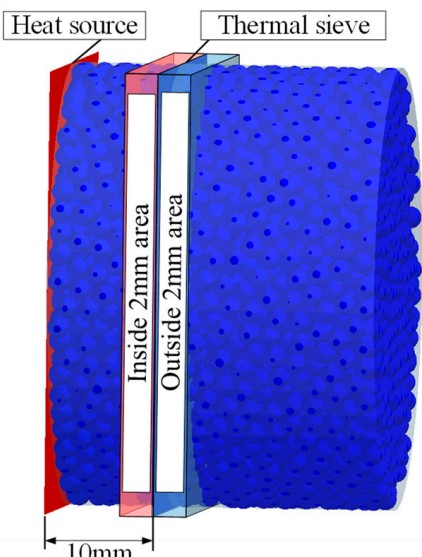

**Figure 2.** Parameter-matching simulation model.

The thermal conductivity and specific heat capacity of the sieve are matched based on the average temperature difference $T_d$ within the range of 2 mm on the inner and outer sides of the sieve, and the average temperature $T_{in}$ of particles within the range of 2 mm on the inner side of the sieve.

$$T_d = T_{in} - T_{out} \tag{3}$$

$$T_{in} = \frac{\sum\limits_{i=1}^{n} T_i}{n} \tag{4}$$

$$T_{out} = \frac{\sum\limits_{i=1}^{m} T_i}{m} \tag{5}$$

where $T_d$ is the temperature difference, $T_{in}$ is the average temperature of the particles within the range of 2 mm on the inner side of the thermal sieve, $T_{out}$ is the average temperature of the particles within the range of 2 mm on the outer side of the thermal sieve, $T_i$ is the temperature of particle $i$, $n$ is the number of particles within the inside 2 mm area, and m is the number of particles within the outside 2 mm area.

*3.2. Design of Simulation Test Group*

In this study, the simulation test design is based on the central composite design method [21]. Central composite design is one of the response surface methods that can obtain approximate relationships between various factors and test results using fewer tests. The simulation test matrix is presented in Tables 3 and 4, which show the specific heat capacity range of the thermal sieve to be 351 to 2048 J/kg °C, and the thermal conductivity range of the thermal sieve to be 17 to 582 W/m K. Coded values refer to the representation of true values in code, while non-coded values represent true values. The code value 0 represents the center point of the parameter value. The code values +a, −a indicate high and low values that affect parameter design. A group of simulation tests are set up for comparison without the thermal sieve, denoted by W0.

**Table 3.** Simulation test matrix.

| Test Number | Specific Heat Capacity of Thermal Sieve (J/kg °C) | Thermal Conductivity of Thermal Sieve (W/m K) |
|---|---|---|
| W1 | 1200 | 17 |
| W2 | 1200 | 300 |
| W3 | 2048 | 300 |
| W4 | 1200 | 300 |
| W5 | 1800 | 100 |
| W6 | 600 | 500 |
| W7 | 1800 | 500 |
| W8 | 1200 | 582 |
| W9 | 351 | 300 |
| W10 | 600 | 100 |
| W11 | 1200 | 300 |

**Table 4.** Coded values and non-coded values of two factors of the thermal sieve.

| Code Value (Horizontal) | Non-Coded Value (Actual Value) | |
|---|---|---|
| | Specific Heat Capacity of Thermal Sieve (J/kg °C) | Thermal Conductivity of Thermal Sieve (W/m K) |
| −a | 351 | 17 |
| −1 | 600 | 100 |
| 0 | 1200 | 300 |
| +1 | 1800 | 500 |
| +a | 2048 | 582 |

## 4. Analysis of Simulation Matching Test Results

Figure 3 displays the simulation test results. The abscissa denotes the distance from the heat source, and the ordinate denotes the average temperature at the distance. The particle temperature variation trend is consistent outside the 2 mm area. The addition of a thermal sieve has a significant effect on temperature change near the sieve, as depicted in Figures 4 and 5. Design points represent data points for simulation. The upper surface design point indicates that the simulation result is higher than the fitting result, that is, the data points are above the fitting surface. The lower surface design point indicates that the simulation result is lower than the fitting result, below the fitting surface. The dotted line in Figure 5 represents a 95% confidence interval.

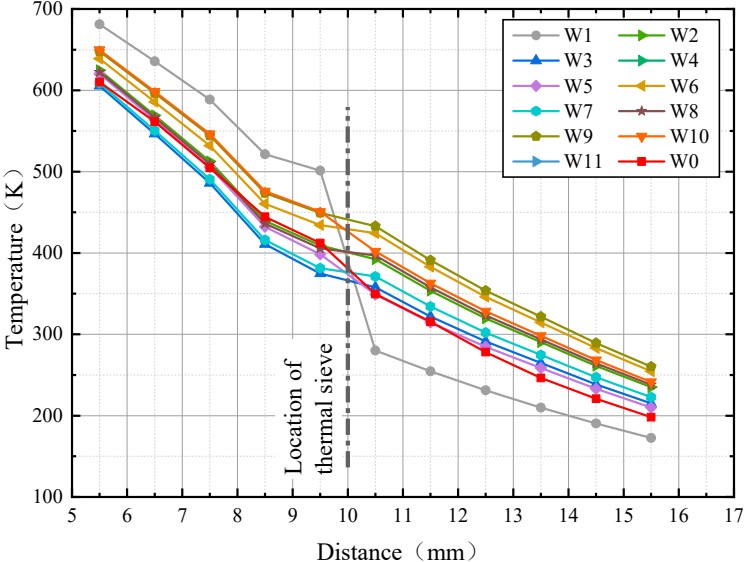

**Figure 3.** Temperature change data.

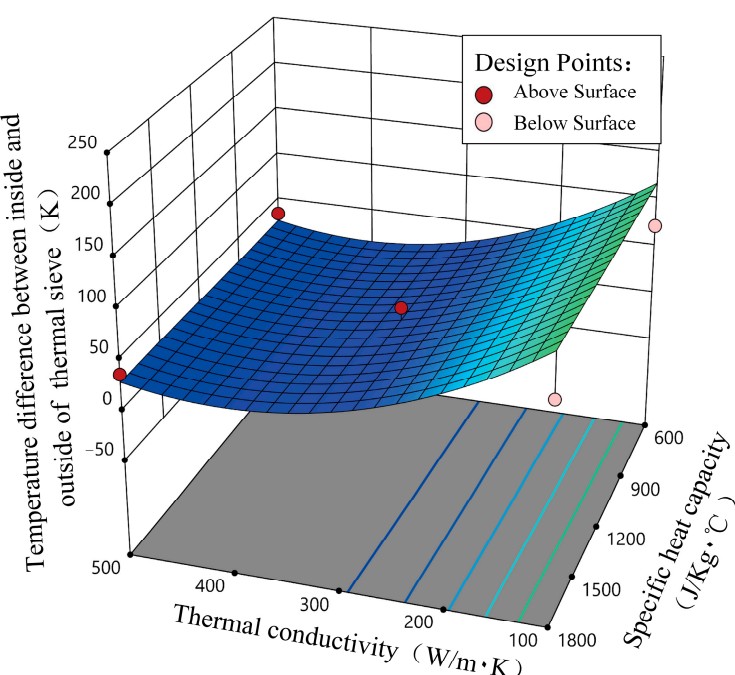

**Figure 4.** The trend of how the thermal conductivity and specific heat capacity of the thermal sieve affect the temperature difference inside and outside the sieve.

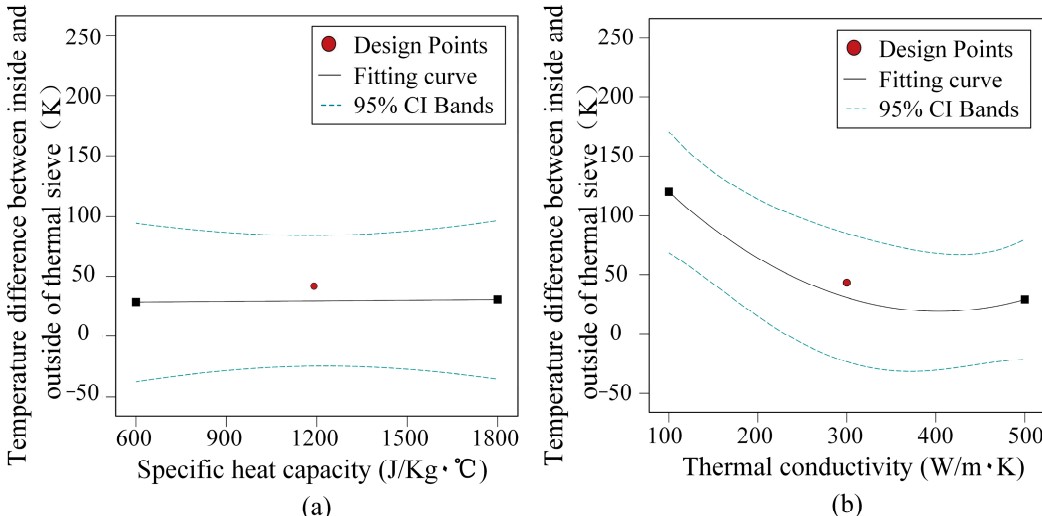

**Figure 5.** The influence of single factor on the temperature difference in the area of 2 mm inside and outside the sieve; (**a**) the influence of specific heat capacity; (**b**) the influence of thermal conductivity.

The temperature difference, $T_d$, between the inside and outside of the thermal sieve, initially decreases and then increases as the heat conductivity of the sieve increases. This relationship is largely independent of the specific heat capacity of the sieve. To determine the relationship between $T_d$ and the parameters of the thermal sieve, the test results were fitted using a quadratic regression equation with experimental design software (Design-Expert):

$$T_d = 195.03231 + 0.001741 C_{Wall} - 0.883485 \lambda_{Wall} + 0.001094 \lambda_{Wall}^2 \tag{6}$$

where $C_{wall}$ is the specific heat capacity of thermal sieve, and $\lambda_{wall}$ is the thermal conductivity of thermal sieve.

The specific heat capacity has little influence on the temperature difference, so it is not considered when matching the thermal conductivity of the sieve. According to the W0

group test data, the $T_d$ is about 92.4 K without the sieve, and the specific heat capacity of the sieve is set at 1200 J/kg °C temporarily. Using Formula (6), the thermal conductivity of the sieve is estimated to be about 144 W/m K.

As seen in Figures 6 and 7, the temperature inside the sieve, $T_{in}$, decreases with an increase in the specific heat capacity of the sieve, and then increases with an increase in the thermal conductivity. Since the relationship between $T_{in}$ and $C_{Wall}$ is mainly linear, only the first term of $C_{wall}$ is considered when fitting the test results with the regression equation. The relationship between $T_{in}$ and the sieve's parameters is determined as follows:

$$T_{in} = 539.60656 - 0.041123 C_{Wall} - 0.383554 \lambda_{Wall} + 0.000471 \lambda_{Wall}^2 \tag{7}$$

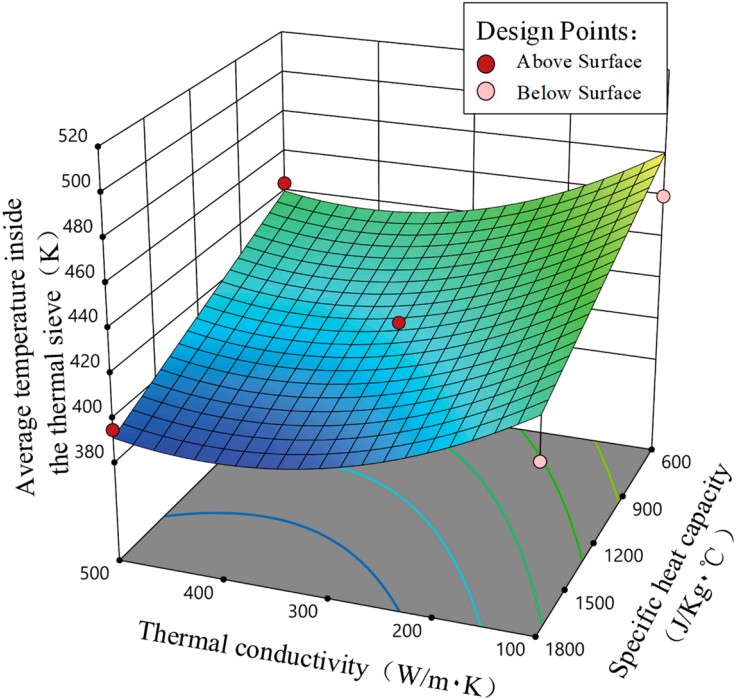

**Figure 6.** The influence the thermal conductivity and specific heat capacity of the thermal sieve on the average temperature of the 2 mm area.

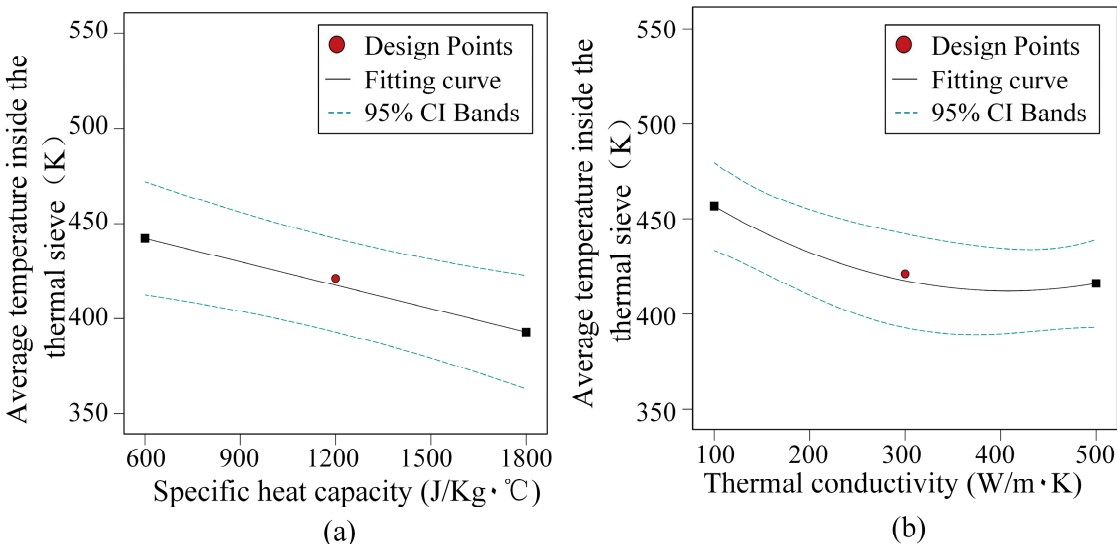

**Figure 7.** The influence of single factor on the average temperature of the 2 mm area inside the thermal sieve; (**a**) the influence of specific heat capacity; (**b**) the influence of thermal conductivity.

To match the specific heat capacity of the thermal sieve, $T_{in}$ is determined at the time at which the thermal sieve is not added, which is approximately 446.8 K. Based on the thermal conductivity of the thermal sieve $\lambda_{Wall}$, which is determined by the temperature difference $T_d$ and found to be around 144 W/m·K, the specific heat capacity of the thermal sieve $C_{Wall}$ is calculated using Formula (7), and is estimated to be approximately 1605 J/kg °C.

## 5. Verification and Analysis of the Drilling Test of the Lunar Icy Regolith

In order to verify the effectiveness of the thermal sieve model, a drilling experiment was conducted under a simulated lunar environment using a section sampling drilling tool efficiency experiment rig (Figure 8). Temperature data of the simulated lunar soil during drilling were collected using a temperature sensor. Discrete element simulation of the drilling was performed for lunar icy regolith with and without a thermal sieve, and the temperature change data of particles in the chip removal area were analyzed. The chip represents the particles discharged by the drilling tool during the drilling process. By comparing the experimental data with the simulation data, the accuracy of the model of the lunar icy regolith after adding a thermal sieve was verified.

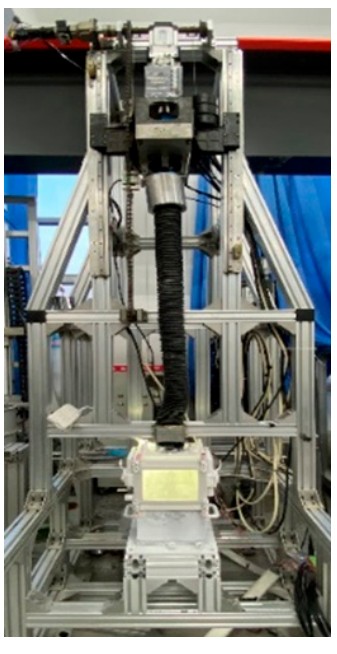

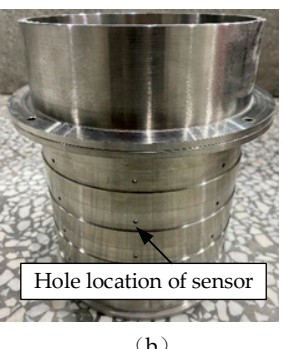

（b）

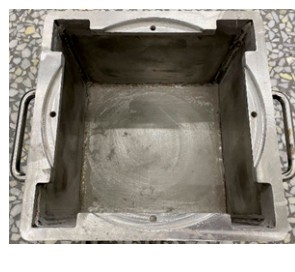

（a）             （c）

**Figure 8.** Experiment device; (**a**) experiment device; (**b**) lunar soil barrel; (**c**) liquid nitrogen drum.

### 5.1. Experiment Equipment and Process

The drilling experiment equipment is depicted in Figure 8a. The temperature sensor is positioned at the hole of the side wall of the lunar soil barrel, as shown in Figure 8b. The liquid nitrogen barrel shown in Figure 8c is used to maintain the low temperature environment of the drilling area by using nitrogen and to prevent external factors from affecting the experiment. The inner diameter of the lunar soil barrel is 70 mm, and the depth is 150 mm, which is consistent with the simulated lunar soil barrel size. The soil barrel temperature sensors for this experiment month are arranged in two layers, with four sensors arranged on each layer. The layout of the temperature sensors is shown in Figure 9. The numerical value in Figure 9b shows the distance between the sensor temperature measurement point and the surface of the drilling tool. One sensor is also positioned in the chip removal accumulation area to measure the temperature change of the chip removal. The experiment process is shown in Figure 10.

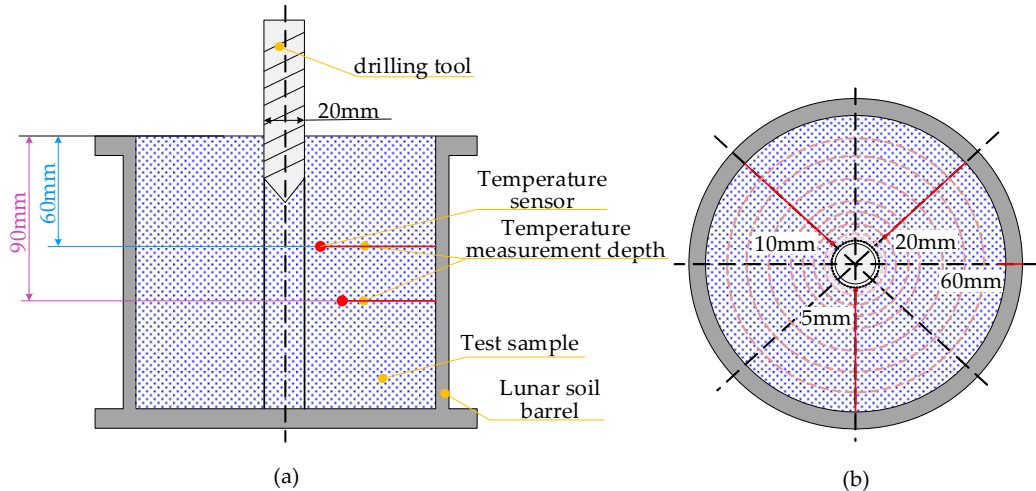

**Figure 9.** Temperature sensor arrangement; (**a**) section sensing arrangement; (**b**) sensor layout per layer.

**Figure 10.** Experiment procedure.

After drying plagioclase and basalt of various particle sizes, prepare them according to Table 5 and mix them evenly [22]. After adding 5 wt% of liquid water, use a press to compact the sample to a density that is substantially consistent with that of the real lunar soil. Insert a temperature sensor, and then cool the sample in an environment of $-80\,°C$ for more than 8 h to ensure that the liquid water is completely frozen. After that, put the prepared sample into the nitrogen box of the experimental platform, and inject liquid nitrogen to cool the sample to $-180\,°C$. When the ambient temperature and sample temperature are basically stable, lower the drilling tool so it is flush with the sample surface. When the temperature of the drilling tool drops to about $-100\,°C$, start drilling. The average feed speed $v_1$ of the drilling tool is 0.5 mm/s, and the rotation speed $n_2$ is 200 rpm. During drilling, the temperature of the cuttings is collected through a temperature sensor arranged above the sample.

**Table 5.** Types and proportions of dry soil used in the experiment.

| Mineral Category | Particle Size | Percentage Content |
|---|---|---|
| Plagioclase | 0.025–0.05 mm | 31.568% |
| | 0.05–0.075 mm | 6.797% |
| | 0.075–0.1 mm | 10.545% |
| | 0.25–0.5 mm | 10.545% |
| | 0.5–1 mm | 10.545% |
| Basalt | 0.025–0.05 mm | 13.502% |
| | 0.05–0.075 mm | 2.920% |
| | 0.075–0.1 mm | 4.526% |
| | 0.25–0.5 mm | 4.526% |
| | 0.5–1 mm | 4.526% |

*5.2. Simulation Process*

A multi-stage particle size simulation drilling model with a thermal sieve was created. The drill tool size was set to be the same as that used in the experiment. The initial temperature of the drill tool was based on its temperature before the test. The simulated lunar soil bucket and thermal sieve were set up as shown in Figure 1b.

The simulation process was divided into two steps. The first step was to create particle samples in the software, and the second step was to perform a drilling simulation in the software. In the first step, particles of different sizes were first generated in two regions. Small particles were generated in the first region, while large particles were generated in the second region. The initial positions of particles appearing in each region were random, and the generation process of particles was dynamic. Due to the action of gravity, the generated particles gradually deposited. In order to obtain a dense sample, we set the friction coefficient between particles to 0 in this step. The small particles and large particles did not flow with each other due to the thermal sieves. After the particles completely settled, the drilling simulation began. The specific parameters of particles are shown in Tables 1 and 2. The initial temperature of the particles and the thermal sieve was set to $-196\,°C$, and the temperature of the lunar soil barrel was also $-196\,°C$, and insulated. The penetration rate and rotary speed were set to be the same as in the experiment. Another multi-stage particle size simulation drilling model was also created without a thermal sieve, with all other parameters being the same. After the drilling was complete, the average temperature of the temperature measurement area in the chip removal zone was calculated for different drilling depths. The temperature measurement area for chip removal is shown in Figure 11.

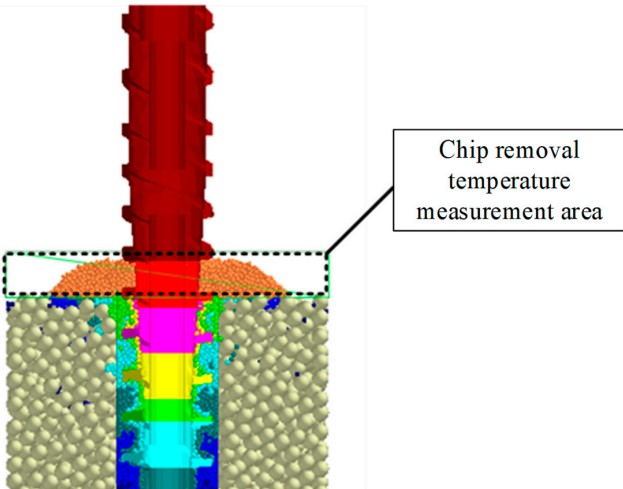

**Figure 11.** Chip removal temperature measurement area.

*5.3. Result Aanalysis*

Figure 12 illustrates the temperature distribution when the drilling depth is 100 mm. Figure 12a displays the case without thermal sieve. In this case, the particles in the first and second areas are mixed. In comparation with Figure 12b, there are fewer particles in the temperature measurement area of chip removal, and the particle temperature is significantly lower. The state of the debris dump in Figure 12b is basically the same as that generated by drilling the simulated lunar icy regolith [23]. As the icy regolith in the drilling area is constrained by the outer icy regolith, it is discharged upwards during actual drilling. The movement limitation on particles in the drilling area, caused by the thermal sieve, may make the simulation closer to the actual drilling and better reflect the temperature changes in the particles. Therefore, Figure 12b is more consistent with actual drilling.

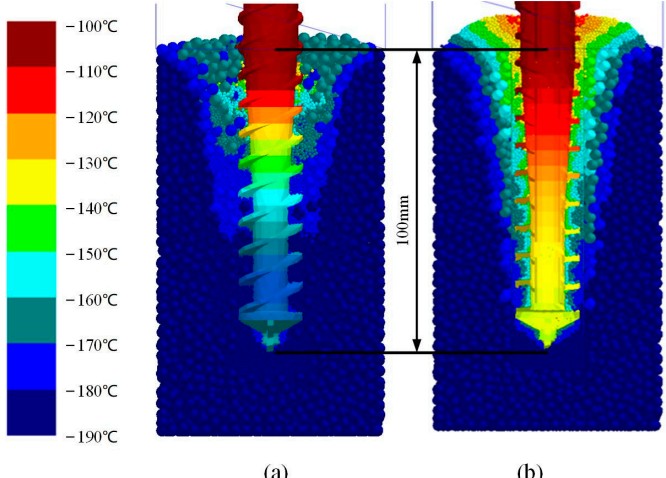

(a)                    (b)

**Figure 12.** Distribution of simulated temperature field; (**a**) Without thermal sieve; (**b**) Added thermal sieve.

Figure 13a shows the radial temperature distribution at a depth of 25 mm, when drilling to 75 mm and 100 mm. The X-axis represents the distance from the center of the drilling hole, the Y-axis represents the temperature corresponding to the distance, and the shaded red rectangle represents the drilling position. It can be seen that the temperature when drilling to 100 mm is generally higher than that when drilling to 75 mm, but the temperature distribution trends of both are basically the same. The temperature difference is mainly due to the longer drilling time as the drilling depth increases, resulting in an increase in the heating time at the same position. Figure 13b shows the radial temperature

distribution at different times at a hole depth of 50 cm, obtained by Formisano through a finite element analysis [24]. The temperature distribution trends of both are the same. As the heating time increases, the temperature increases.

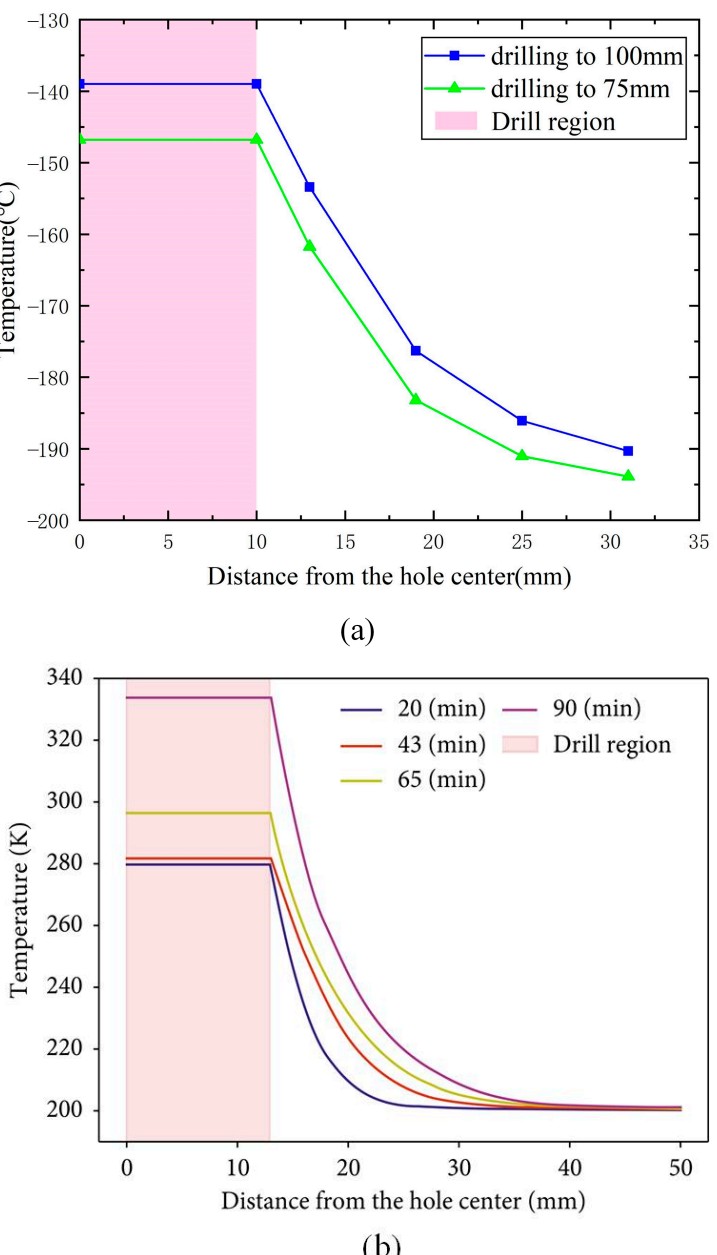

(a)

(b)

**Figure 13.** Radial temperature distribution; (**a**) discrete element model simulation data with added thermal sieve; (**b**) finite element model simulation data established by Formisano (Formisano, 2022).

As shown in Figure 14, the temperature variation trend in the chip removal area measured in the experiment is almost the same as that in the simulated chip removal area. After the drilling depth exceeds 40 mm, the simulated temperature data without thermal sieves stop rising, and the data fluctuate significantly. This is in poor agreement with the experimental data. After adding the thermal sieve, when the drilling depth exceeds 20 mm, the temperature shows a stable upward trend, which is consistent with the experimental data trend. The temperature of the chip removal area simulated with the addition of the thermal sieve is higher than that simulated without the addition of a thermal sieve, and the difference between the two continues to increase after the drilling depth exceeds 40 mm. At the initial stage of the experiment, there was a temperature

difference of 20 °C from the initial set temperature of the simulation. After removing the influence of the initial temperature difference, when drilling at a depth of 100 mm, the temperature deviation between the experimental chip removal area and the simulated chip removal area after adding a thermal sieve is about 5 °C, and the temperature deviation between the simulated chip removal area without heating the sieve is about 10 °C. When the drilling depth is around 50 mm, the experimental temperature value begins to decrease, because the experimental data are fixed point temperature values measured by a single temperature sensor, and not the average temperature of the chip removal area. Therefore, it is possible for the temperature sensor to come into contact with some low-temperature chip removal, causing certain errors. This is mainly due to the impact of measurement methods. The revised experimental data (dashed line) in Figure 14b represent the temperature data after eliminating the impact of the initial temperature difference and the temperature measurement method. After eliminating this impact, the simulation data after adding the thermal sieve are in good agreement with the experiment data.

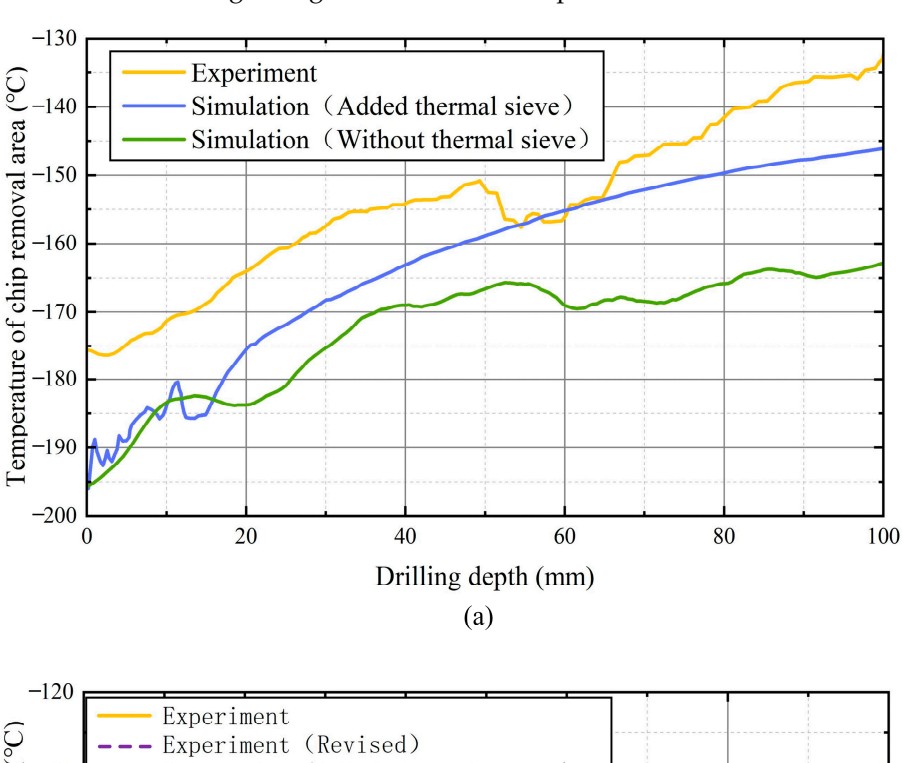

(a)

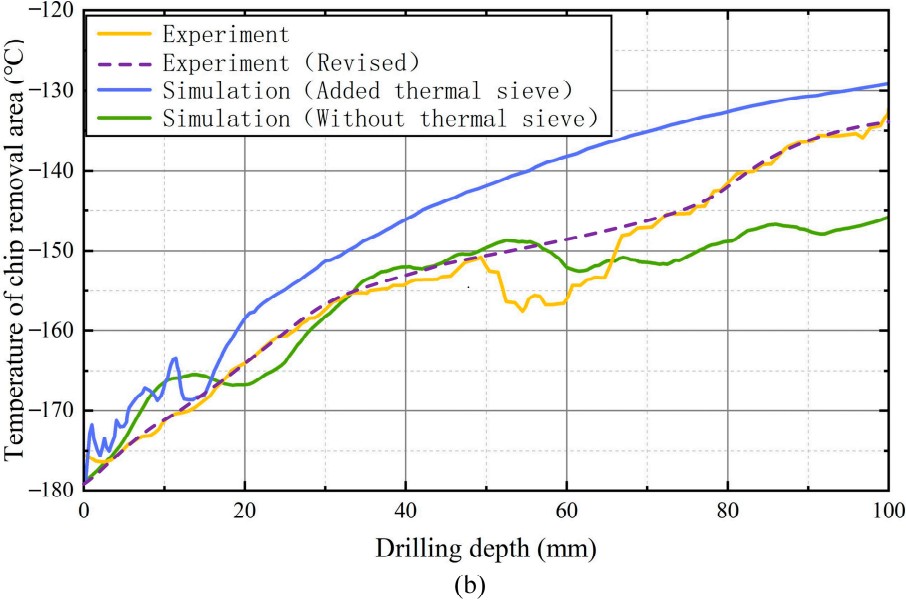

(b)

**Figure 14.** Temperature of chip removal area; (**a**) raw data for experiments and simulations; (**b**) data after eliminating the impact of initial temperature difference and measurement method.

To verify whether the multi-stage particle size simulation with a thermal sieve can reduce the computation time, this study includes an additional simulation model in which the entire region is filled with small particles. This model does not include a thermal sieve, and all other parameters remain the same. The computing platform used for this study has an Intel i9 12,900 K central processing unit and an RTX3080 graphics processing unit. Since using a single small particle size simulation takes too long, the predicted time for the single particle size filling simulation is based on the estimated time provided by the discrete element software, which is 363.02 h. The predicted time for the multi-stage particle size simulation with a thermal sieve is 50.675 h. The simulation time for the multi-stage particle size simulation with thermal sieve is estimated to be reduced by approximately 86% compared to the simulation model filled with a single particle size.

## 6. Conclusions

This paper proposes an optimization method for a multi-stage particle size simulation model by adding a thermal sieve to limit the flow between particles in different areas. The results show that the model with a thermal sieve can effectively restrict particle flow and ensure chip removal during drilling, with an average temperature of the chip removal area that closely follows the temperature trend measured in the experiment. The temperature deviation is $\pm 10\,^{\circ}\text{C}$. Additionally, the calculation time is reduced by about 86% compared to the single small particle size filling model. Therefore, when the computational power is limited, adding a thermal sieve to the multistage particle size simulation model means the temperature increase caused by particle flow in different regions can be effectively avoided. This method can be used in DEM simulations to reasonably predict temperature changes in the debris removal area during the process of drilling lunar icy regolith.

**Author Contributions:** Conceptualization, D.Z. and W.Z.; methodology, D.Z.; software, Z.C. and H.W.; validation, Z.C., J.C. and W.Z.; formal analysis, Z.C. and H.W.; investigation, Z.C.; resources, W.Z.; data curation, W.Z., J.C. and Z.C.; writing—original draft preparation, D.Z., Z.C. and H.W.; writing—review and editing, D.Z., W.Z. and J.C.; visualization, Z.C.; supervision, W.Z.; project administration, D.Z. and W.Z.; funding acquisition, D.Z. and W.Z. All authors have read and agreed to the published version of the manuscript.

**Funding:** This project was financially supported by the fundamental research funds of the National Natural Science Foundation of China (No. 51805488, No. 52005136), and the Science and Technology Program of Guangzhou (No. 202102020320).

**Data Availability Statement:** The data that support the findings of this study are available from the corresponding authors upon reasonable request.

**Conflicts of Interest:** The authors declare no conflict of interest. The funders had no role in the design of the study; in the collection, analyses, or interpretation of data; in the writing of the manuscript; or in the decision to publish the results.

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
