# Peer review of "Numerical Modeling of Thermal Behavior during Lunar Soil Drilling"

_aerospace, doi:10.3390/aerospace10050472_

Round 1

Reviewer 1 Report

The manuscript investigates the utilization of DEM in lunar thermal modelling of drilling operations. It proposes the use of a thermal sieve method to reduce computational time in such research. Although this has multiple merits for future researchers, I would strongly recommend improving the manuscript in the topical area of drilling simulation.

More effort needs to be given to further describe the model in the coupling of mechanical interactions of the drill with particles, in particular the methodology and results described in Figures 10 and 11. It would be of great importance to the readers to better understand the verification method and its results that were described in the experimental test procedure.

It would also be beneficial to understand how phase change of water influences the thermal analysis in the model and experimental results (or might influence the results if it has not been implemented).

Overall, sections 5 and 6 of the manuscript need further elaboration and discussion of the results. Multiple researchers have previously investigated the problem of planetary drilling in dry and icy regoliths, and their influence on the thermal behaviour of the subsurface, but none of them are referenced nor their results discussed and compared in this manuscript. This must be improved.

Author Response

Dear Reviewer 1

We would like to thank the reviewer for conducting such a detailed review of the article. The comments are valuable and helpful for revising and improving our article, as well as the important guidance to our research. We have made revisions to the manuscript based on the reviewer's comments and provided a detailed response in the attachment.

Reviewer 2 Report

Dear Authors,

The subject of this paper is relevant to the journal and the approach combining numerical simulations with experimental measurements is commendable. However, before being considered for publication, I recommend addressing the following major concerns:

- Please expand on the explanation of the thermal sieve and how it relates to grain size. The fact that particle flow is to be restricted by such a sieve is difficult to relate to the action of drilling. Figure 1 does not provide enough detail to understand the thermal sieve, nor does it specify the grain sizes involved (despite what the reference to the Figure on l.139 says). l.154-155 mention that there are 10 thermal sieves set up, which contributes to the overall uncertainty about what a sieve is. If there are 10 sieves, please show these in Figure 1. As the thermal sieve is really key to the point of the paper, it is difficult to assess the readiness for publication as long as this is not clarified.

- Following that first point, it is difficult to understand the purpose and mechanics of Section 3. It may become apparent once the point above is clarified. Notes: Table 3 does not appear in its entirety; Table 4: what are coded and non-coded values? What does "a" represent?

- Section 4 presenting results is overall difficult to assess too, due to a general lack of clarity. Admittedly, addressing point 1 above might clarify a number of things, but Figures 4 to 7 need more information in the caption. In particular, what are above and below surface design points in Figures 4 and 6, as well as the design points in Figure 5 and 7? What do the dashed lines represent in Figures 5 and 7?

- Section 5.1: Please describe how samples are prepared. From Figure 9 it seems that liquid water is first added? This has an important influence on the nature of the sample (solid vs. granular) and needs to be detailed.

- Section 5.2: The simulation process is not well defined. Please add some wording to clarify, for example, why small and large particles concentrate in either portion of the thermal sieve. Are particles broken down into finer ones? This would be key to understand both how the simulation works and the results presented, and is yet not specified.

- There is a general lack of referencing and justification of choices/assumptions. Examples include:
    o Equation 1: please reference papers/books where this equation was found
    o Section 2.1: where can one read up on this previous work? If this has not been published yet, it needs to be marked as personal/private communication and explained in more detail.
    o When introducing the effective heat conductivity, references would be welcome.
    o Section 2.2: In order to assume conduction is dominant, one would have to verify that vapor pressures generated by the drilling remain low to not contribute to heat conduction by convection. While Figure 12 shows temperatures that indicate this is indeed the case, a justification for this assumption needs to be included here.
    o Reference needed for the Hertz-Mindlin contact model.
    o l.127: Which relevant scholars are meant? Please reference them.
    o ALL parameters introduced in Table 1 need to be either accompanied by a reference or justified, ideally both. How were all these numbers chosen?
    o Section 2.3.2: The work by Cui Jinsheng seems to be playing an important role here. It requires a reference. Has this work been published?

- The statement on l.145-147 needs to be justified. 1. The heating of small particles is considered instantaneous in this study, as mentioned on l.98. This may no longer apply to larger particles and the time needed for heating a set of large particles may be much longer than the time needed to heat a set of small particles. 2. Larger particle samples are usually also more porous (although this may not be the case due to the particles being spherical). Radiative transfer may become more prominent and larger portions of larger particles are exposed rather than in contact with other particles. These two factors need to be addressed and their potential irrelevance justified.

- l.303-305: The error analysis is extremely hand-wavy and does not support the end statements of the paper. This needs to be developed for a credible conclusion.

- Section 5.3, simulation time analysis: A simulation of 363h can in principle be run within ~15 days. Was this done to check out the thermal sieve code? If so, then why are these run times only estimations and not actual data?

- The term "chip" was never properly defined. Its use is thus confusing.

- The last paragraph of the introduction is a summary of the paper rather than an introduction. When all the other issues are addressed, I recommend re-phrasing adequately.

Minor comments and typos:

- l.135: typo in distribution
- l.252: I don't think it is Figure 4 that is meant to be referenced.
- Figure 10 is not referenced in the text.

Author Response

Dear Reviewer 2

We would like to thank the reviewer for conducting such a detailed review of the article. The comments are valuable and helpful for revising and improving our article, as well as the important guidance to our research. We have made revisions to the manuscript based on the reviewer's comments and provided a detailed response in the attachment.

Reviewer 3 Report

Aerospace-2297549_review_comments

This manuscript outlines a technique intended to reduce computational cost of simulating thermal transport during drilling in lunar soil. The authors propose incorporation of  a "thermal sieve" to segregate sizes of mobile particle in the drilling zone. An effective thermal conductivity is assigned to the "sieve" in order to match thermal properties.

The idea is interesting and innovative. The manuscript, however, is exceptionally difficult to understand and is woefully incomplete.

This paper seems to be an extension of Jinsheng Cui's work, but lacks sufficient background explanation. It took several reads before I understood what the authors were attempting to simulate. I am still unsure if there is a comparison of the authors' results to DEM simulations.  Part of this uncertainty lies in the ambiguity of how heat is transferred with the particles as these move. Are the particles moving to locations that they would not otherwise? How does the extra degree of freedom associated with particle temperature affect particle mechanics?

The language is inconsistent throughout - thermal sieve, thermal conductivity sieve, heat transfer sieve, and so on. The term experiment is used for models, simulations, and actual experiments.

The authors fail to cite sources. For example, where did equation 1 come from? What is the source for the particle properties listed in Tables 1 and 2?

Line 145: The authors claim "particle diameter has little influence on the heat transfer between particles", which on the surface is not true. This claim needs elaboration and explanation with supporting evidence since there is a distribution of particle sizes.

Line 180: To what were thermal conductivity and specific heat matched?

Line 186: Why was 2 mm chosen for the temperature averaging?

Line 193: The authors presume readers are intimately familiar with "Central Composite Design" and offer up Tables 3 (incomplete) and 4 without explanation.

Figures 4-7: Are the authors really defining fitting functions with only 3 simulation points?

Line 250: It is unclear in this paragraph as to what is a physical experiment and what is a simulation.

section 5.1: There is little to no details for the physical tests. For example, how was 5% water content established and verified? What are the dimensions of the apparatus? The test procedure illustrated in Figure 9 is not explained. For example, what were the relative locations of the temperature sensors?

Figure 10 is not discussed.

Line 303: The authors claim good agreement between simulated and test data when "taking into account the measurement error", but there is no discussion on measurement error. There is also no explanation as to why the two simulations were started at 20 degrees less than the experiment.

Recommendation is Reject: There may be merit to the thermal sieve approach, but there is insufficient detail and case studies to make any claims as to utility. This was a frustrating manuscript to review.

Author Response

Dear Reviewer 3

We would like to thank the reviewer for conducting such a detailed review of the article. The comments are valuable and helpful for revising and improving our article, as well as the important guidance to our research. We have made revisions to the manuscript based on the reviewer's comments and provided a detailed response in the attachment.

Reviewer 4 Report

1.    Table 3 and Table 4 are incomplete. It got cut during submission. So, it is difficult to make comments on Fig. 3. Why W0, and W10 has similar slope inside the 2 mm area and different from others? Why W1 has completely different behavior?

2.      What is the chamber pressure for sublimation tests?

3.      How authors mixed the ice with the lunar regolith simulant? What type of regolith was used in this work?

4.      Was the drill heated? How was it heated?

5.    In Figure 12, except close to the 60 mm drilling depth area, other drilling depths have significantly higher values for tests as compared to the simulation results. Can authors provide an explanation for this? What are the uncertainties associated with the measurements and initial conditions?

6.  Why does the test data show downward peak near the 60 mm drilling depth that is not visible for the simulation results?

7.      Authors should provide a detailed explanation of the test facility and test procedures.

8.  The simulation and test data are far. Authors also mention the discrepancy is due to the measurement and simulation uncertainties. Thus, which results are acceptable for the scientific community?

9.     In Table 1, particle thermal conductivity (W/m·K) has been used as 14.760. Typically, lunar regolith thermal conductivity is 0001 to 0.0300 W m-1 K-1. Can authors provide the reason for choosing this value?

10.  Table 1 uses particle sizes as 0.6 mm and 2.4 mm. What is the reason for choosing 0.6 mm particle diameter for sublimation simulations where average particle size is ~ 0.07 mm? Moreover, only 10% of the lunar regolith particle size is more than 1 mm. Then why did authors consider 2.4 mm particles? Also, can authors provide the particle size distribution for the experimental regolith simulant?

https://www.lpi.usra.edu/science/kring/lunar_exploration/briefings/lunar_soil_physical_properties.pdf

11.  Line 145, “Since the particle diameter has little influence on the heat transfer between particles, using larger particle diameters in these areas can gradually reduce the number of particles”. Line 97, “When the particles have a small diameter, heat transfer inside the particles can be considered instantaneous, and the temperature distribution within the particle is assumed to be uniform”. So, in this case the larger diameter particles might not have uniform temperature distribution. Does this have any impact on the heat transfer between the particles?

12.  Line 165, “The density of the sieve has little effect on heat transfer and is therefore not matched”. Line 170, “Hence, the thermal conductivity and specific heat capacity of the sieve are the matching parameters in this simulation”. However, the relative heat conduction capability as compared to storage capacity of material depends on thermal diffusivity that also contains the density. Can authors explain why the density of sieve has little effect on heat transfer?

13.  Line 126, “The parameter settings of the simulated lunar soil particles are shown in Table 1 and Table 2, and the reliability of these parameters has been verified by relevant scholars to meet the requirements of the simulation model of the icy lunar soil”. Can authors provide references for the source of these parameters?

14.  Line 34, “However, Through….”. Through should be lowercase.

15.  Line 41, “-70—-60℃”. It would be more readable if “-70 ℃ to - 60 ℃.

16.  Line 41, “The drilling tool will generate a certain amount of heat in the process of drilling and sampling. During the process of drilling and sampling, the drilling tool generates heat, which…”. Both lines have the same meaning.

17.  Line 44, “phase change temperature of water ice and cause vaporization”. This solid-vapor transition should be termed as sublimation instead of vaporization.

18.  Line 71, “lunar ice soil”. It is typically termed lunar icy regolith and regolith simulant.

19.  Line 97, “When the particles have a small diameter, heat transfer inside the particles can be considered instantaneous, and the temperature distribution within the particle is assumed to be uniform”. Line 116, “The temperature inside each individual particle is assumed to be uniformly distributed”. What is the maximum size of the particle that can be assumed to be small for uniform temperature distribution?

20.  Line 95, “However, other forms of heat transfer can be approximated to the effective heat transfer between particles, which is represented by the ETC, denoted as ks”. What are those forms of heat transfer? How are they related to the ks? Is there any equation?

21.  Line 115, “The simulated lunar soil is assumed to be composed of spherical particles”. Is ice also considered spherical? Is ice considered in contact with regolith particles?

22.  Line 125, “Hertz-Mindlin model and parallel bond model are coupled to simulate the contact force of the lunar soil”. Is there any reason to select the parallel bond model?

Author Response

Dear Reviewer 4

We would like to thank the reviewer for conducting such a detailed review of the article. The comments are valuable and helpful for revising and improving our article, as well as the important guidance to our research. We have made revisions to the manuscript based on the reviewer's comments and provided a detailed response in the attachment.

Round 2

Reviewer 1 Report

As stipulated in the previous comments, research needs to address and be more transparent:

1. With reference to simulating drilling and coupling mechanical and thermal behaviours there is no visibility of both experimental and model foundations of the issue.

2. Although computational time conclusions might be a good byproduct of the research, the topical "Numerical Modeling of Thermal Behavior during Lunar Soil Drilling" which aims to "investigate the temperature changes that occur in the debris removal area during the drilling process" would need more verification, discussion, and conclusions in these areas. I do not see any improvements so far.

3. The discussion and quality of references are subpar. I cannot imagine that an old Zacny paper with a spoil geometry discussion can constitute state-of-the-art research. What about data we have from recent studies, both R&D and even flight model development (like VIPER or ExoMars drills)?

Author Response

Dear Reviewer 1
We apologize for not achieving the modifications that satisfy you. We would like to thank you for conducting such a detailed review of this paper once again. These comments are valuable and helpful for the revision and improvement of our paper, as well as important guidance for our research. We have made revisions to the manuscript based on your feedback and provided a detailed response in the attachment.

Reviewer 2 Report

Dear Authors,

Thanks for addressing my review points. I recognize good progress in the paper, but still have the following comments to be worked on:

- Figure 1 is much better now. It would be of interest to show the drill in it too, in order to understand how far from the sieve it is.

- The rationale for the thermal sieve is still not clear. It is now understood that it supports the proper transition between a region with small particles and a region with larger particles, and it is also clear why a region with larger particles is of interest. However, would there normally be particle transfers between these two regions when drilling? Also, why does the sieve have to have a thermal conductivity? Is it not possible to directly transfer heat from small particles to large ones without a sieve?

- Please provide at least one reference for central composite design.

- Section 5.2: Particle generation. I am assuming that particle generation is happening before the drilling simulation starts. If correct, please specify that in the text. From your answer to my Point 5, it sounds like particles would be able to flow during particle generation. Is that the case? In general, a few more words on how exactly these particles are placed in the simulation volume would help understanding the phrasing currently used.

- Point 7: I see you opted to just remove the sentence in question from the paper. I think the answer you are providing to me in the cover letter has good wording. Please consider including it into the paper.

- Error analysis: It would really support the "good agreement" statement if the error was shown in Figure 13. I understand that the offset is quantified now, but the error itself is not really. It is just mentioned as "measurement error". How much is the measurement error and if you offset the blue and gree curves in Figure 13 by the 20°C mentioned and add a shaded area around the curves to show the measurement errors, one could see indeed that the agreement is good (and maybe better for the green than for the blue curve).

Author Response

Dear Reviewer 2
We would like to thank you for conducting such a detailed review of this paper once again. These comments are valuable and helpful for the revision and improvement of our paper. We have made revisions to the manuscript based on your feedback and provided a detailed response in the attachment.

Reviewer 3 Report

Overall, the manuscript is significantly improved in readability and clarity. There are still a few items that stand out.

As part of the author's response to the 1st review comment, they explain that the "drilling area is constrained by the outer icy regolith" and "the added motion restriction [from the thermal sieve] helps better reflect the temperature change at the particles".

Two points of clarification are needed here:
1. Does this mean that the thermal seive improves particle motion in the model, or is this only true for icy regolith?
2. Why is this not mentioned in the manuscript?

It is still not clear if the parameter matching must be done for every new setup up, or if the limited testing is sufficient modeling a change in experiment conditions.

With regard to the 12th review comment, the line numbers for section 5.3 are 383-404 in the manuscript available from Aerospace, not 345-358 as discussed by the authors. The authors' explanation, also added to the manuscript does not address measurement error. I think there is some confusion between measurement error and the difference between test conditions and simulation conditions. It is not a measurement error if measured temperature decreases at drilling depth of 60 mm due to conditions occurring during testing. There is a discrepancy between simulated and experimental conditions that should be explained, but this cannot be categorized as "experimental error".

Author Response

Dear Reviewer 3
We would like to thank you for conducting such a detailed review of this paper once again. These comments are valuable and helpful for the revision and improvement of our paper. We have made revisions to the manuscript based on your feedback and provided a detailed response in the attachment.

Reviewer 4 Report

The reviewer is satisfied with the responses. Thank you.

Author Response

Dear Reviewer 4
We would like to thank you for your recognition of this paper. Your comments are valuable and helpful in revising and improving our paper.

Round 3

Reviewer 2 Report

Dear Authors,

Thanks for your answers and clarifications, I think the paper is much better in its present form. A few more points still:

- Point 2 of previous review: I understand the purpose of the sieve now. I strongly recommend that you specify in the text of the paper the details you have provided in your review answer. Indeed, as you are concentrating on the thermal properties of particles, it seems you are not implementing any of their other mechanical properties, such as cohesion, as well as sliding and rolling friction, which would impeed them from moving outwards radially. This is an important point to specify in order to understand the usefulness of the sieve.

- Figure 13 and l.357-369: The conclusion "the thermal model established through discrete element method can reasonably predict the temperature changes during the drilling process." is not supported by the Figure and the text. Indeed parts a and b of the Figure show different things and scales are not the same. All one can conclude as it is right now, is that the trends are consistent in both the DEM and FEA.

- Figure 14 and associated text: Please provide a bit more detail about the graph in the caption. In particular, it is useful to describe the offset adjustments you have performed. This is included in the text, but should be mentioned in the figure caption.
l.386-388: It is not 100% clear what is meant here. Is the theoretical curve generated from removing the offset and then the measurement errors from the simulation data? In that case, naming it "theoretical" is misleading. If that is not what it is, then please reword to clarify.
l.382: It seems the experiment temperature value decreases already at a drilling depth of 50 mm, rather than 60 mm, fro the looks of in Figure 14.
l.380-381: I think you need to specify that for drilling depths between 20 mm and ~80 mm, the simulation without the thermal sieve seems to yield better results than the one with the thermal sieve. This trend is only reverse from 80 to 100 mm.

Author Response

Dear Reviewer 2

First of all, thank you for your recognition of our work. Thank you for conducting such a detailed review of this paper again. We have made revisions to the manuscript based on your feedback and provided a detailed response in the attachment.
